# β-Glucans Could Be Adjuvants for SARS-CoV-2 Virus Vaccines (COVID-19)

**DOI:** 10.3390/ijerph182312636

**Published:** 2021-11-30

**Authors:** Alfredo Córdova-Martínez, Alberto Caballero-García, Enrique Roche, David C. Noriega

**Affiliations:** 1Department of Biochemistry, Molecular Biology and Physiology, Faculty of Health Sciences, GIR: “Physical Exercise and Ageing”, University Campus “Los Pajaritos”, Valladolid University, 42004 Soria, Spain; 2Department of Anatomy and Radiology, Faculty of Health Sciences, GIR: “Physical Exercise and Ageing”, University Campus “Los Pajaritos”, Valladolid University, 42004 Soria, Spain; alberto.caballero@uva.es; 3Department of Applied Biology-Nutrition, Institute of Bioengineering, Miguel Hernández University, 03202 Elche, Spain; eroche@umh.es; 4Instituto de Investigación Sanitaria y Biomédica de Alicante (ISABIAL), 03010 Alicante, Spain; 5CIBER Fisiopatología de la Obesidad y Nutrición (CIBEROBN), Instituto de Salud Carlos III (ISCIII), 28029 Madrid, Spain; 6Spine Unit, Department of Surgery, Ophthalmology, Otorhinolaryngology and Physiotherapy, Faculty of Medicine, Hospital Clínico Universitario de Valladolid, 47003 Valladolid, Spain; noriega1970@icloud.com

**Keywords:** β-glucans, COVID-19, immunomodulators, SARS-CoV-2, vaccination

## Abstract

Waiting for an effective treatment against the SARS-CoV-2 virus (the cause of COVID-19), the current alternatives include prevention and the use of vaccines. At the moment, vaccination is the most effective strategy in the fight against pandemic. Vaccines can be administered with different natural biological products (adjuvants) with immunomodulating properties. Adjuvants can be taken orally, complementing vaccine action. Adjuvant compounds could play a key role in alleviating the symptoms of the disease, as well as in enhancing vaccine action. Adjuvants also contribute to an effective immune response and can enhance the protective effect of vaccines in immunocompromised individuals such as the elderly. Adjuvants must not produce adverse effects, toxicity, or any other symptoms that could alter immune system function. Vaccine adjuvants are substances of wide varying chemical structure that are used to boost the immune response against a simultaneously administered antigen. Glucans could work as adjuvants due to their immunomodulatory biological activity. In this respect, β-(1,3)-(1,6) glucans are considered the most effective and safe according to the list issued by the European Commission. Only glucans with a β-(1,3) bond linked to a β-(1,6) are considered modulators of certain biological responses. The aim of this review is to present the possible effects of β-glucans as adjuvants in the efficacy of vaccines against SARS-CoV-2 virus.

## 1. Introduction

COVID-19 is caused by the SARS-COV-2 virus. Although most patients undergo mild symptoms, 20% develop severe symptoms. The disease is characterised by a primary pneumonia that evolves to acute respiratory distress syndrome (ARDS) in the most severe cases [1]. Despite the huge number of studies on SARS-CoV-2, there is no conclusive data regarding the best treatment for the disease [2]. Therefore, a treatment approved by the relevant regulatory health systems does not yet exist. Awaiting for the appearance of an effective treatment (single or combined), the current alternative is prevention and the use of vaccines against the virus.

Aside from the debate regarding the types and doses to be used to achieve the goal of immunisation, vaccination is the most effective strategy in the fight against the pandemic [3]. The main objective of preventive vaccination is the induction of a specific immune response against a pathogenic microorganism leading to protection against the infection and in the long term, eradication of the disease [4,5]. mRNA vaccines, such as those developed against COVID-19, work by instructing the cells of the host to synthesize an antigen that elicits an immune response to block or destroy the virus [6]. Other SARS-CoV-2 vaccines are viral proteins, protein fragments, or vectors containing an inactivated virus version [7].

Different natural biological products exerting immunomodulatory actions can be administered with vaccines or taken orally. These are known as adjuvants and can work by alleviating the symptoms of COVID-19 as well as enhancing the action of the vaccine itself. Vaccine adjuvants are substances of a wide chemical structure that are used to boost the immune response against a specific antigen. In this context, glucans could be considered as adjuvants due to their immunomodulatory biological activity [8,9]. Adjuvants also contribute to an effective immune response not only at an early age, when the immune system is still immature and vaccination is necessary, but they can also enhance the protective effect of vaccines in immunocompromised individuals including elders [10,11]. Therefore, adjuvants are used for several purposes: (a) As essential components or complements of vaccines; (b) to stimulate antibody production; (c) as tools to decrease the inflammatory response; or (d) to evaluate hypersensitivity responses in toxicological assays. A variety of compounds have been used as potential vaccine adjuvants, including minerals, microbial products, emulsions, saponins, cytokines, polymers, microparticles, and liposomes [12]. 

According to Edelman [13], adjuvants can be of three types: (a) active immunostimulants, (b) carrier proteins, and (c) vehicle-type adjuvants. Therefore, the use of adjuvants in the vaccination process may be of great interest because: (a) they can increase the specific immune response (seroconversion) and achieve effective protection against various diseases; (b) they can facilitate the use of smaller amounts of antigen in a vaccine and thus achieve a higher coverage in the population; (c) they can achieve a more rapid protection of the population, reducing virus spread; (d) they can reduce vaccine doses, improving vaccine administration, reducing logistics and improving the cost/benefit ratio; and (e) they can target the immune response to achieve more effective mechanisms against disease.

Adjuvants must not produce adverse effects, toxicity, or any other symptoms that could affect the function of the immune system. In this respect, β-(1,3)-(1,6)-glucans are considered safe according to the list issued by the European Commission (SANCO/12826/2011 Rev. 2 (POOL/E4/2011/12826/12826R2- EN.doc). The list establishes the general health claims allowed on foods, including those related to the reduction of disease risk and to children’s development and health. Only glucans with a β-(1,3) bond linked to a β-(1,6) bond are considered to be biological modifiers of the immune response.

The aim of the present review is to present in detail the possible immunomodulatory effects of β-glucans and their possible use as oral adjuvants for vaccines against SARS-CoV-2. Glucans as biological response modifiers could be a complementary strategy to vaccination. Oral β-glucans have been described as prophylactic supplements to boost the immune response and to attenuate the symptoms of COVID-19 [14]. The use of these compounds could be a key element by aiding in the efficacy of vaccines against SARS-CoV2, and thus be accurate in controlling the COVID-19 pandemic.

Moreover, it has been found that vaccine effectiveness against SARS-CoV-2 infection decreased with time after vaccination, from 88% the first month to 47% after five months. This decrease seems to be due to a decline in vaccine efficacy rather than an increased virulence of SARS-CoV-2 variants [15]. Therefore, it is necessary to produce an optimal immune response with vaccination. Using adjuvant molecules in parallel would be an interesting strategy, in order to obtain a better immune response and even reduce the vaccine dose, and to improve the efficiency and efficacy of the vaccine. As indicated, adjuvants can raise immunogenicity or weaker immunogens, by increasing the effect of vaccines and reducing the amount of antigen and the frequency of immunisation required for protective immunity [16].

## 2. Systemic Impact of COVID-19 Infection

The immune response is crucial against SARS-CoV-2 infection. In general, the immune response against viruses takes place in two phases: A first non-specific (innate immune) phase that starts immediately after virus entry into the host, and a second specific (late) phase known as adaptive or acquired immunity [17].

COVID-19 presents as an exacerbated immune response against SARS-CoV-2, resulting in systemic damage accompanied by inflammation [18,19] that triggers a cytokine overproduction termed as “cytokine storm”. This particular process plays a key role in the pathogenesis and severity of the disease [20].

The onset of COVID-19 infection leads to an increase in the expression of proinflammatory cytokines, such as interleukin (IL)-1β, IL-2, IL-6, IL-8, tumour necrosis factor-α (TNF-α), and interferon (IFN) α and β, among others. The increase observed in these cytokines correlates to the severity of the infection [21]. In summary, the overproduction and expression of cytokines reflects the systemic damage caused by infection and may act as adaptive signalling in the response against SARS-CoV-2 [22,23,24,25]. This entire response is undoubtedly exacerbated by the stress component of the situation, evidenced by glucocorticoid synthesis that has a direct impact on the immune system [26].

Peripheral blood mononuclear cells include different types of lymphocytes (T cells, B cells and NK cells), dendritic cells (DCs), and monocytes [27]. DCs are antigen-presenting cells for T lymphocytes [28]. As COVID-19 infection progresses, neutrophil and lymphocyte depletion (neutropenia and lymphopenia) occur in severe stages [29,30,31]. SARS-CoV-2 invasion and/or replication could be favoured by increased apoptotic events on T-lymphocytes and the binding of SARS-CoV-2 spike protein to CD147, although this last point needs further investigation [22]. When disease severity increases, both the number and function of monocytes and DCs are altered [32]. DCs express MHC class I and II antigen-presenting molecules and are functionally the most potent inducers of antigen-presenting T cell activation and proliferation [33]. In addition, the glycan mannose present in many pathogens is recognised by DC-specific intracellular adhesion molecules (ICAMs) [34]. Therefore, an important element for the control of SARS-CoV-2 infection could be the inhibition and dissemination of pathogens in DCs.

In addition, NK cells act through membrane receptors to trigger the release of the hydrolytic/cytolytic content from their granules. The control of infection by NKs consists in the destruction of virus-infected cells, leading to a restriction of viral replication [35,36,37]. The production of certain cytokines, such as IFNs, IL-2, and IL-12, are associated with NK cell action [38]. IFNs activity increases the cytolytic capacity of active NK cells which in turn leads to IL-2 and IL-12 expression [39,40]. IFNs interfere with the use of host cell resources by the virus, preventing replication and provoking cell death (apoptosis) [41]. In addition, it has been observed that NK cell depletion may favour pulmonary complications that occur in the most severe cases of COVID-19 [42]. Moreover, in models of respiratory infection, activated NK cells can deteriorate lung injury [43]. This suggests that therapeutic measures need to improve and enhance NK cell functionality. This could be a therapeutic alternative in severe cases of COVID-19 [43].

## 3. The Cytokine Response during COVID-19 Infection

In general, a key point to control the inflammatory process is the neutralisation of circulating TNF-α to membrane receptors [44]. We have observed this phenomenon when supplementing athletes with an immunomodulator during post-exercise recovery [44,45,46,47,48]. The neutralisation of pro-inflammatory cytokines (mainly TNF-α) and intermediates (nitric oxide/NO and hydrogen peroxide/H_2_O_2_) is key to modulate inflammation, although an alternative to block TNF-α is the use of antibodies against it [49]. Most drugs also block bioavailable TNF-α, increasing susceptibility to co-infection or re-infection [50]. In addition, TNF-α can be released from macrophages via a β-glucan-mediated mechanisms. Lentinan (a β-glucan extracted from the mushroom *Lentinula edodes*) enhances the cytotoxicity of peritoneal macrophages against metastatic tumours [51]. Lentinan structure consists in β-1,3-linked glucose units that every five residues have two β-1,6-linked side chains. On the other hand, lentinan can cause inhibition of TNF-α and NO production in murine macrophages, through the inhibition of inducible nitric oxide synthase (iNOS) gene expression as well as TNF-α mRNA [52]. 

So far, it seems that the most relevant antiviral defence is the action of IFNs. They recruit neutrophils, modulate inflammation and thus infection. Therefore, IFNs are placed as the first barrier against invasion by SARSCoV-2 [53,54]. In these circumstances, IFNs would be able to block virus dispersion, giving the organism time to mount a more potent and specific immune response against SARS-CoV-2 [55,56]. In the early stages of viral infection, IFN-I (α/β) act directly on NK cells by activating and enhancing cytotoxic function [57,58]. It has been observed that patients with severe COVID-19 infection have lower IFN expression, which is associated to a decrease in the differentiation of CD4+ and CD8+ lymphocytes and NK cells [59]. In addition, high IL-6/IFN ratio could be a predictor of disease severity and thereby of lung damage associated to the “cytokine storm” [57,60]. In this context of infection, structural and non-structural SARS-CoV2 proteins inhibit IFN-I release and secretion [61,62,63]. This blocks the antiviral and immunomodulatory activity of IFN-I, decreasing resistance to infection. Some of the adjuvant properties of glucans (commented later) against SARS-CoV-2 may lie along these lines of action.

## 4. β-Glucans

β-glucans can be obtained from mould fungi, yeasts such as *Saccharomyces cerevisiae*, mushrooms such as *Lentinus edodes*, and some types of algae. Other sources include cereals, such as oats and barley, and to a lesser extent rye and wheat [64]. β-glucans are glucose polysaccharides forming part of the cellular wall of certain pathogenic fungi and bacteria. There is a wide variety of β-glucans, which differ in the position of the carbon bonds between the different monomers forming the glycoside chain. Bonds (1→3), (1→4), (1→6) have been identified between the D-glucose units. Depending on the source of extraction, one type of polysaccharide or another predominates in the β-glucans [65,66]. Insoluble β-glucans (1,3 and 1,6) have the highest biological activity when compared to their counterparts soluble β-glucans (1,3 and 1,4). Therefore, differences between bonds and chemical structure in relation to solubility seem to be instrumental for the mode of action and biological activity [67,68].

### 4.1. Initial Immunomodulatory Effect of β-Glucans

In general, glucans are a group of substances with immunomodulatory biological activity and have been successfully used in clinical medicine and shown to improve health [8,9]. Certain β-glucans have immunomodulatory effects, intervening in both innate and adaptive immunity. Basically, the proposed mechanisms to exert these effects are as follows (Figure 1): (a) After oral administration, β-glucans are absorbed in the intestine and recognized by intestinal macrophages, which degrade them into smaller polysaccharides. They are then transported and released into the medulla and the mononuclear phagocytic system [69]. Subsequently, fragments of the β-glucans are released by macrophages and captured by neutrophils (granulocytes), monocytes and DCs, triggering the immune response; (b) An alternative mechanism of β-glucan uptake is through recognition by epithelial cells, such as M cells, cells of the lymphoid tissue associated with the intestine, macrophages, and DCs, introducing them into circulation [70].

### 4.2. Receptors for β-Glucans

From an immunological point of view, the innate immune system must display the capability to recognise and respond quickly to a pathogen. On the other hand, adaptive immunity uses generated receptors that recognize antigens that the host has previously encountered. The β-glucans are recognised by immune system cells via the following receptors: Dectin-1, CR3, Lactosylceramide, selected scavenger receptors, and the Toll like receptor (TLR) (Figure 1). The affinity of β-glucans for these receptors depends on their molecular weight, having higher affinity for receptors with high molecular weights [71].

After the uptake and release of β-glucans in the marrow and mononuclear phagocyte system, they are recognised by granulocytes, monocytes, or macrophages and NK cells via receptors such as CR3. The binding of the polysaccharide to the receptor triggers different responses, such as phagocytosis and/or cytokine secretion (Figure 1), thus activating innate immunity [72]. In addition, Dectin-1 (or b-glucan receptor) is a type II transmembrane protein receptor that binds β-(1,3) and β-(1,6) glucans. Dectin-1 binds β-glucans and triggers innate immune responses, including phagocytosis and the production of pro-inflammatory factors, directed towards the elimination of infectious agents [73,74,75].

Monocytes/macrophages and neutrophils express Dectin-1 as well as DCs and a subpopulation of T cells, but to a lesser extent [76]. Dectin-1 binds polysaccharides containing β-(1,3) and/or β-(1,6) bonds. The signalling pathways activated by Dectin-1 involve the participation of NF-κB, the signaling adaptor protein CARD9, and nuclear factor of activated T cells NFAT [77,78,79]. This activation process culminates with the secretion of IL-6, IL-10, IL-12, and TNF-α. 

The Dectin-1 receptor works together with TLR2 in macrophages and DCs, inducing the production of pro-inflammatory cytokines [80]. TLRs do not recognise glucans by themselves but enhance glucan signalling from Dectin-1 binding, resulting in NF-κB activation and cytokine production. Signalling from both receptors results in the phosphorylation of the ITAM (immunoreceptor tyrosine-based activation motif), leading to phagocytosis and activation of NADPH oxidase, ensuing the death of the invader microbe [81]. In addition, the SIGN-related 1 (SIGNR1) of DCs containing the ITAM-3 homologue motif, is another major macrophage mannose receptor that cooperates with Dectin-1 in the recognition of β-glucans and subsequent phagocytosis [82].

On the other hand, TLR-4 blockade can inhibit the production of IL-10 and IL-12 induced by purified glucans from Ganoderma. This suggests a key role of TLR-4 signalling in glucan-induced DC maturation. This effect is also mediated by increased IκB kinase, NF-κB activity, and MAPK phosphorylation [83].

Moreover, the adaptive immune response requires the combined action of T-cells and antigen-presenting cells. Presentation of MHC (major histocompatibility complex)-I antigens to cytotoxic CD8+ T cells is restricted to peptide fragments coming from intracellular pathogens after proteasome activity. The MHC-II endocytic pathway presents only proteolytic peptides from extracellular pathogens to CD4+ T-helper cells. β-glucans can use the MHC-II pathway to activate CD4+ T-cells [84]. In this context, β-glucans are processed into smaller carbohydrates that bind to MHC-II on antigen-presenting cells (DCs) for presentation to T helper cells.

Another mechanism of action of β-glucans is mediated by activated complement receptor 3 (CR3), located on NK cells, neutrophils, monocytes and certain lymphocytes. This pathway accounts for opsonic recognition of β-glucans, leading to phagocytosis and lysis of reactor cells. The β-glucans bind to the lectin domain of CR3, priming the binding of inactivated complement 3b (iC3b) on the surface of reactor cells. Neutrophils containing CR3 can trigger lysis of iC3b-coated tumour cells [72]. In a similar way, most human NK cells express CR3, and opsonisation of iC3b-coated NK cells has been shown to lead to increased lysis of the target [85,86].

### 4.3. Immunostimulatory Activity of β-Glucans

It has been well stablished that macrophages play a key role in all phases of host defence in case of infection through a complete innate and adaptive immune response. Binding of the β-glucan to the receptor triggers different responses: phagocytosis and/or cytokine secretion, thus activating the innate immune system [69,72]. Macrophages are also involved in the secretion of cytokines (IL-1, IL-6, IL-8, IL-12, TNF-α) and inflammatory mediators (NO and H_2_O_2_) (Figure 2). Thus, the activation of macrophage functions by β-glucans can enhance host immune defence. Among the cytokines secreted, TNF-α, recognised as the first cytokine produced by activated macrophages, is an essential molecule in host defence. TNF-α activates inflammatory reactions by triggering the secretion of ILs, including IL-1 and IL-6. TNF-α increases NO release and activates innate and acquired immune responses, increasing immunoglobulin (Ig) production, activating the complement pathway and producing T and B cells. TNF-α also has an impact on neutrophil action [87].

The binding of β-glucans to Dectin-1 elicits the secretion of cytokines such as IL-6, IL-10, IL-12, and TNF-α, which increases the body’s response against antigens [67]. Certain studies have shown that some β-glucans are able to activate the complement system through the complement system receptor CR3, also known as CD11b/CD18. CR3 is expressed in myeloid cells, NK cells, and lymphocytes [69,72,88]. Once the β-glucans are recognised by CR3, the complement C3 pathway is activated, and the C3b fraction acts directly on macrophages, NK cells, and neutrophils, increasing phagocytosis and lysis of cells that are coated with iC3b (inactivated C3b) [69,72,88]. Therefore, β-glucans, through their action on different receptors (Dectin-1 and CR3) of the immune system, can trigger a variety of responses. These responses affect cells of the immune system, including macrophages, neutrophils, monocytes, NK cells, and DCs [89].

### 4.4. β-Glucans against Infection

β-glucans may have broad anti-infective effects. In fact, β-glucan has been found to have a protective effect against influenza virus [90]. In swine influenza virus (SIV)-induced pneumonia, β-glucans induced significantly higher levels of IFN-γ and NO in bronchoalveolar lavage fluid of treated pigs compared to the untreated group [90]. Another study in rats showed that treatment with β-glucans resulted in increased neutrophil migration to the site of inflammation and improved antimicrobial function [91]. Moreover, in patients undergoing major surgery, administration of β-glucans improves monocyte and neutrophil function, playing a key role in decreasing infectious complications [92]. Similarly, peri-operative administration of β-glucans also reduces by 39% severe post-operative infections [93]. Moreover, the β-glucan extract has recently been reported to show in vitro immunomodulatory and pulmonary cytoprotective effects, being indicated for COVID-19 immunotherapy [94].

### 4.5. β-Glucans as Immunomodulators

The key to immunomodulation is to adapt the host response. Otherwise said, how can the response be augmented or suppressed to obtain an optimal immune response? The advantages are: (i) enabling host defence mechanisms by enhancing the immune response and (ii) not involving the use of antibiotics, which is important taking into account the large number of pathogens that have developed resistance to antimicrobial agents [95].

There are several types of endogenous immunomodulators, such as IFN-γ, colony stimulating factor (CSF), and macrophage and monocyte stimulating factor (GM-CSF). Some polysaccharides isolated from various botanical sources appear to possess relevant immunomodulatory properties and relatively low toxicity [81,96]. As mentioned before, it has been observed that β-glucans can increase the cytotoxic activity of macrophages against tumour cells and microorganisms. This protective action is achieved by increasing reactive oxygen species (ROS) and NO production, as well as enhancing the production of cytokines and chemokines, such as TNF-α, IL-1β, IL-6, IL-8, IL-12, IFN-γ, and IFN-β2 [9,25,97].

The role of glucans as immunomodulators has long been known. Glucans represent a type of immunostimulatory active molecules across the evolutionary spectrum, representing a conserved alternative in the direct innate immune response against pathogenic fungi [9,97]. Thus, the components of the immune system are activated and modulated by β-glucans. It would therefore appear to be of interest as an ideal element to promote effective and long-lasting immunity.

## 5. Glucans Complementing the Action of Vaccines

A significant property in the immunological activity of β-glucans is their adjuvant action. In general, the immune system reacts with a better and greater response when β-glucans “prime” this system. For this reason, many studies have focused on the use of these compounds in conjunction with viral, bacterial and parasitic vaccines [90].

During immune modulation, β-glucans are recognised through transmembrane receptors. Glucan-receptor binding stimulates glucan phagocytosis, the release of proinflammatory cytokines, chemokines, and antimicrobial proteins. In this context, it has been reported the potential performance of glucans for the preparation of delivery nanosystems as a platform for DNA vaccination [98].

Current vaccines against SARS-CoV-2 are from Pfizer-BioNTech (SARS-CoV-2 mRNA vaccine), Moderna (SARS-CoV-2 mRNA vaccine), Janssen Pharmaceutical Companies of Johnson & Johnson (J&J’s) (based on a replication-incompetent adenovirus serotype 26 (Ad26) vector, which encodes a SARS-CoV-2 spike protein), and Oxford/AstraZeneca (made from virus ChAdOx1) [99,100,101].

Studies exploring the protective effects of β-glucans and its possible use as adjuvants in vaccines were conducted in the early 1990s. The β-glucan, like the tuberculosis vaccine (BCG), induces a TRIpartite Motif (TRIM) phenotype, involved in pathogen recognition and transcriptional regulation of innate immunity systems [102,103,104]. β-Glucans can produce long-lasting TRIM against a wide range of pathogens. In addition, β-glucans show additive and even synergistic effects with various agents [105]. The ability of (1→3), (1→6)-β-glucans to activate cellular and humoral components of the immune system is well known. They enhance the antimicrobial activity of mononuclear cells and neutrophils and increase the function of macrophages. In addition, they stimulate monocyte and macrophage proliferation [95]. Upon exposure to β-glucans, innate immune cells are activated and there is an increase in cytokine production accompanied by changes in metabolic functions [106,107]. Subsequently, if these cells are confronted with heterologous secondary stimuli, they are reprogrammed to produce a more potent immune response [108,109,110] (Figure 2). 

As mentioned before, β-glucans have antiviral effects, helping to decrease the severity of infections. After exposure to an initial stimulus, the antiviral response could be due to the induction of immunity through TRIM, which triggers metabolic, mitochondrial, and epigenetic reprogramming, resulting in a memory phenotype of enhanced immune response [8,90,111,112].

Therefore, immunomodulators appear to be an effective tool to combat infectious diseases by boosting host defences. These can be used either preventively through vaccination or therapeutically. Currently, more and more vaccines are showing greater efficacy due in part to adjuvants. These are molecules that have little immunogenicity by themselves but help to boost and direct the immune response against an antigen [113]. In 1997, Adams, et al. [114] suggested that PGG-glucan (Betafectin), obtained from yeast, modulates the production of pro-inflammatory cytokines by lymphocytes and monocytes during sepsis. In studies in mice, it has been observed that those treated with PGG-glucan, when lymphocytes were removed, produced lower amounts of proinflammatory cytokines such as TNF-α. In addition, glucan treatment increased the level of cell apoptosis. Finally, rats infected with a lipopolysaccharide (LPS from *E. coli*) and treated with LAM (β-polysaccharide-(1→3)) displayed a decrease in TNF-α, nitrite and monocytes compared to non-treated rats. LAM also modulated intrahepatic immune cells, decreasing peroxidase-positive cells (corresponding to monocytes/neutrophils) and increasing the number of ED2-positive cells (mainly macrophages) [115].

## 6. Conclusions

β-glucans activate the immune system and modify the cellular response. The binding of β-glucans to their receptors can elicit a cellular response through modulation of the activities of several extracellular messengers, such as cytokines and chemokines [116]. We believe that β-glucans orally administered may be useful as adjuvants, improving the effectiveness of various vaccines currently marketed against SARS-CoV-2 [117]. β-glucans show a clear immunomodulatory effect by improving the cellular pattern involved in the infectious process and decreasing the cytokines involved in the inflammatory course derived from COVID-19. The improvement of immune function is key in the mechanism of protection and elimination of intracellular infectious agents. Thus, oral β-glucans are promising molecules as vaccine adjuvants, as they alone can stimulate various immune pathways, including antibody production. β-glucans enhance the immunogenicity of hepatitis B vaccine, influenza vaccine, and vaccines against systemic aspergillosis and coccidioidomycosis. Thus, β-glucans (particularly β-(1,3)-(1,6)) are potentially useful as immune adjuvants to enhance the immune response, as has recently been observed in the vaccine against avian influenza subtype H5, with no adverse reactions [118].

## Figures and Tables

**Figure 1 ijerph-18-12636-f001:**
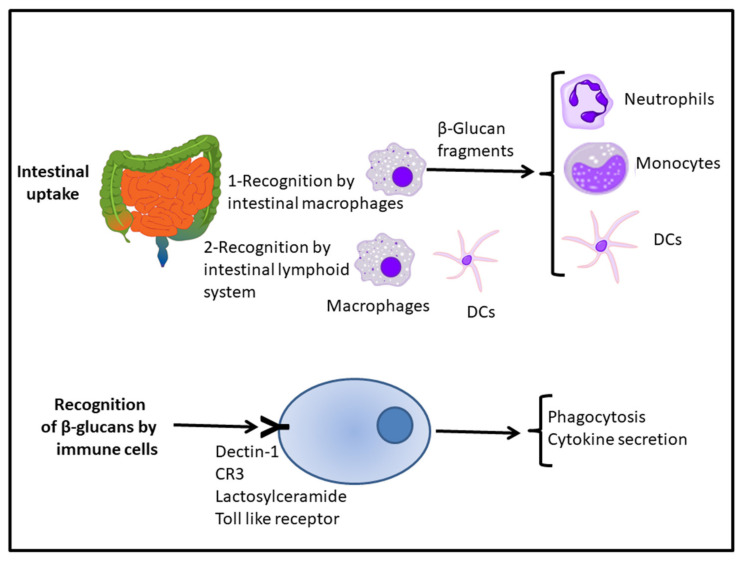
Initial stages in the immunomodulatory action of β-glucans. Scheme representing the initial steps during the intestinal uptake and the subsequent interaction with receptors located in immune cells. See text for more details. Abbreviations used: DCs, dendritic cells. Images modified from Wikimedia commons.

**Figure 2 ijerph-18-12636-f002:**
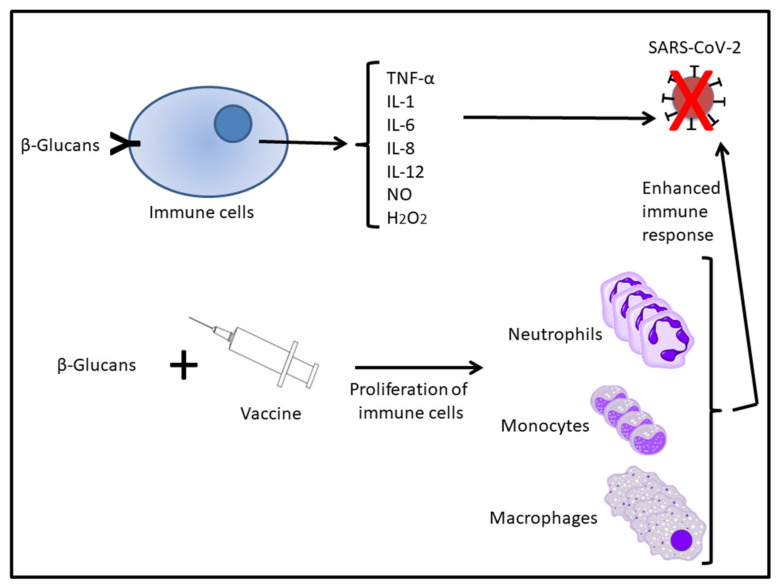
Immunomodulatory action of β-glucans alone and in combination with vaccines. See text for more details. Images modified from Wikimedia commons.

## Data Availability

Not applicable.

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
