# Peer review of "β-Glucans Could Be Adjuvants for SARS-CoV-2 Virus Vaccines (COVID-19)"

_ijerph, 2021, doi:10.3390/ijerph182312636_

Round 1
Reviewer 1 Report
The presented review “Glucnas [β-(1,3)-(1,6)] as Adjuvants for Vaccines against 2SARS-CoV-2A Virus (COVID-19)” is discussing about the glucans and their immunomodulatory effects in vaccines preparation. However, article did not discuss beta glucans role or their presence in covid vaccines. It is unclear that whether these vaccines are contains this component if yes how much they needed in proportion to the covid antigens? Overall, the title is not truly reflecting what the article contained. Either edit the title or include the beta glucans presence in covid vaccines if there is any.
Author Response
REVIEWER-1
The presented review “Glucans [β-(1,3)-(1,6)] as Adjuvants for Vaccines against 2SARS-CoV-2A Virus (COVID-19)” is discussing about the glucans and their immunomodulatory effects in vaccines preparation. However, article did not discuss beta glucans role or their presence in covid vaccines. It is unclear that whether these vaccines contain this component. If yes how much they needed in proportion to the covid antigens? Overall, the title is not truly reflecting what the article contained. Either edit the title or include the beta glucans presence in covid vaccines if there is any.
ANSWER: The efficiency of β-glucans as adjuvants has been demonstrated when they are taken orally. We have taken into account this particular aspect and changed in the text accordingly. In addition, we have completed the information regarding β-glucans and highlighted in the text in red colour.
Reviewer 2 Report
The authors tried to review β-glucans as a vaccine adjuvant. The biggest problem of this manuscript is that the authors mixed up couple of different things into one story.
- It looks like the authors discussing Beta Glucans as oral supplement to induce cytokines, chemokines aa well as antiviral effect.
- However, if the authors want to discuss Glucans as a vaccine adjuvant, it must be mixed with antigen as a component of vaccine.
- Furthermore, if only β-(1,3) (1,6) Glucans has an activity, the authors should focus to this compound only.
- The authors mentioned “Neutralisation of pro-inflammatory cytokines (IL-6 and TNF-α) are key to modulate inflammation (line 132)” and later “activation of macrophage functions by β-glucans can enhance host immune defence by the secretion of cytokines (IL-1, IL-6, IL-8, IL-12, TNF-α) and inflammatory mediators (lines 234-237)”. As such the story does not flow well.
Minor points:
- Title: What does “A” stand for after SARS-CoV-2?
- Figure 2 showing nothing specific to SARS-CoV-2.
Author Response
REVIEWER-2
The authors tried to review β-glucans as a vaccine adjuvant. The biggest problem of this manuscript is that the authors mixed up couple of different things into one story.
ANSWER
Thank you very much for your comments. We believe we have understood your appreciations and we have taken all this into account, and consequently the text has been modified. Also, we have completed information regarding β-glucans. All this has been highlighted in the text with red colour.
QUESTIONS
- It looks like the authors discussing Beta Glucans as oral supplement to induce cytokines, chemokines as well as antiviral effect.
ANSWER: Yes, we want to propose this line of research to our scientific colleagues: β-glucans as oral adjuvants to complement the action of vaccines against COVID-19. This is supported by the antiviral action that of these compounds.
- However, if the authors want to discuss Glucans as a vaccine adjuvant, it must be mixed with antigen as a component of vaccine.
ANSWER: The efficiency of β-glucans as adjuvants has been demonstrated when they are taken orally. We have taken into account this particular aspect and changed in the text accordingly. In addition, we have completed the information regarding β-glucans and highlighted in the text in red colour.
- Furthermore, if only β-(1,3) (1,6) Glucans has an activity, the authors should focus to this compound only.
ANSWER: We have presented data of β-glucans in general, but in some parts of the manuscript, we have focussed in β-(1,3) (1,6) glucans, because their role as oral adjuvants has been studied more extensively.
- The authors mentioned “Neutralisation of pro-inflammatory cytokines (IL-6 and TNF-α) are key to modulate inflammation (line 132)” and later “activation of macrophage functions by β-glucans can enhance host immune defence by the secretion of cytokines (IL-1, IL-6, IL-8, IL-12, TNF-α) and inflammatory mediators (lines 234-237)”. As such the story does not flow well.
ANSWER: This is a mistake, The only inflammatory cytokine is TNF-α. IL-6 can be pro- or anti-inflammatory cytokine. In any case, we have changed in the text (in red) accordingly.
Minor points:
- Title: What does “A” stand for after SARS-CoV-2?
ANSWER: This a typing error. This has been corrected.
- Figure 2 showing nothing specific to SARS-CoV-2.
ANSWSER: We agree, but editorial team suggest to add some figures to help the reader to understand some basic concepts.
Round 2
Reviewer 1 Report
The manuscript is still lacking the discussion of beta-glucans in COVID vaccines. Whole story around it however it is still lacking the clear discussion of beta glucans components in COVID vaccines.
Author Response
REVIEWER-1
The manuscript is still lacking the discussion of beta-glucans in COVID vaccines. Whole story around it however it is still lacking the clear discussion of beta glucans components in COVID vaccines.
ANSWER:
Dear reviewer, the following is a detailed response to your suggestion. When we raised the article it was not intended, as there are no data, about the effect of β-glucans on SARS-CoV-2.
Therefore, the aim of the present review is to present in detail the possible immunomodulatory effects of β-glucans and their possible use as oral adjuvants for vaccines against SARS-CoV-2. We cannot discuss the use of β-glucans as adjuvants in COVID-19 vaccines, because they are not used actually in the clinical practice.
This review is intended as a proposal for the possible usefulness of β-glucans as enhancers of the effect of the vaccine itself. In addition, it is intended to increase the interest of colleagues in research in this interesting field.
This review pretends to be a proposal to increase the interest of colleagues to investigate in this interesting field.
For this reason, we presented a lot of data regarding the key role of adjuvants in vaccines and the particular action of β-glucans in this respect.
In this context, we introduced adjuvants in lines 55-101. An example of β-glucans as adjuvants is presented in lines 151-158.
Section 4 describes in detail the physiological action of β-glucans in the immune response.
Section 5 defines the role of β-glucans as candidate oral adjuvants of COVID-19 vaccines, taking into account that they are used already as adjuvants with certain vaccines.
The main idea with this information is to justify the use of β-glucans to complement orally the effect of vaccines against SARS-CoV-2
We believe that all this can be a stimulus for researchers and pharmaceutical companies to initiate and invest in this particular line of research.
Reviewer 2 Report
Well revised. The story is clear now.
Author Response
Thank you very much